# Salaried Workers’ Self-Perceived Health and Psychosocial Risk in Guayaquil, Ecuador

**DOI:** 10.3390/ijerph17239099

**Published:** 2020-12-06

**Authors:** Antonio Ramón Gómez-García, Cecilia Alexandra Portalanza-Chavarría, Christian Arturo Arias-Ulloa, César Eduardo Espinoza-Samaniego

**Affiliations:** Esai Business School, Universidad Espíritu Santo, Samborondón 091650, Ecuador; aportalanza@uees.edu.ec (C.A.P.-C.); carias@uees.edu.ec (C.A.A.-U.); ceespinoza@uees.edu.ec (C.E.E.-S.)

**Keywords:** self-perceived health, psychosocial risks, working conditions, salaried workers, Ecuador

## Abstract

Self-perceived health is an important indicator of occupational health. This research explored the relationship between poor self-perceived health and exposure to psychosocial risk factors, taking into account potential socio-demographic, occupational, and employment determinants. Using data from the First Survey of Occupational Safety and Health Conditions, covering 1049 salaried workers in Guayaquil, Ecuador, descriptive and stratified binary logistic regression analyses (odds ratios with corresponding 95% confidence intervals) were carried out. A significant relationship was found between exposure to psychosocial risk factors and the probability of presenting poor self-perceived health by socio-demographic, occupational, and employment characteristics. Occupational exposure factors to psychosocial risks were predictors of self-perceived ill health and were related to the variables analyzed; the most frequently expressed factors among the respondents were cognitive demands (DCOG) and job insecurity (IL). The results have implications in terms of designing effective workplace interventions pursuant to ensuring the health and well-being of employees.

## 1. Introduction

Due to its relative validity and reliability, self-perceived health status (SHS) is one of the most widely used indicators for measuring health in the working population [1,2]. This indicator’s importance lays in its close relationship with workers’ social, economic, and occupational characteristics [3].

Recent research has demonstrated relationships between psychosocial risk factors present in working conditions and negative effects on SHS [4,5], especially in low- and middle-income countries [6,7]. It is important to emphasize that self-perceived ill health in the workplace is multifactorial because it is associated with an organization’s social relationships, work characteristics, and cultural aspects [7,8,9,10,11].

Interpersonal conflicts between colleagues and because of inadequate support from employers have been identified as generating work stress, resulting in poor sleep quality, anxiety states, and depression [12]. Further, concerns about potentially losing current employment manifest themselves in a variety of physical and psychological ailments which, at times, can be so severe that recovery is not possible [13,14]. The greater the demands or requirements of work activities—emotional, cognitive, and quantitative—and the lesser the possibility of control, the greater the likelihood of illness resulting from psychological stress [15,16,17]. Moreover, there is a positive association between long working hours and the risk of suffering from cardiovascular diseases [14], as well as the risk of conflicts between employment and family [18,19]. Finally, job insecurity and employment characteristics have been studied as determinants of health inequalities.

In particular, the Republic of Ecuador is among the unequal countries in Latin America and the Caribbean [20,21] and, to our knowledge, there are no studies on the detrimental effects of psychosocial risks on the health of workers in the country, although they have been widely discussed in other contexts [7]. In the absence of evidence on this reality [14], it is crucial to generate knowledge about this phenomenon for decision-making and public interventions based on evidence to improve working conditions [22].

Accordingly, using data from salaried workers in the Ecuadorian city of Guayaquil, the objective of this study was to determine the relationship between self-perceived poor health and exposure to psychosocial risk factors by socio-demographic, labor, and employment characteristics.

## 2. Materials and Methods

### 2.1. Design, Data, and Population

An observational cross-sectional research design was configured based on data from the First Survey of Occupational Safety and Health Conditions (I-ECSST), covering salaried workers over the age of 18 in the Ecuadorian city of Guayaquil (*n* = 1049) in 2017. Further details on the design and validation of the I-ECSST are publicly available [23]. The sample selection and data collection procedures were similar to previous studies conducted in the country’s capital, Quito [24,25].

### 2.2. Analytical Variables

The dependent variable (SHS) was generated from question Q.42 included in the health conditions dimension of the I-ECSST: “How do you consider your health?” The Likert response scale of the question was dichotomized into good self-perceived health (excellent, very good, and good) and poor self-perceived health (fair, poor, and very poor).

The independent variables used were the psychosocial risk factors at work ascertained from questions included in the psychosocial dimension of the survey: (i) social support from colleagues (ASc), (ii) social support from bosses (ASj), (iii) job insecurity (IL), (iv) emotional demand (DEMO), (v) cognitive demand (DCOG), and (vi) quantitative demand (DCUA). As per the foregoing, all of these variables were considered as possible determinants of poor SHS in the working population.

(i)Q.33 “Do you feel supported by your team?”(ii)Q.34 “Do you feel supported by your superiors in your current work?”(iii)Q.35 “Are you afraid of losing your current job?”(iv)Q.32 “Are you exposed to negative feelings, emotions or treatment from others in the course of your work?”(v)Q.30 “Do you have to make a mental effort to do your job?”(vi)Q.31 “Do you have the time required to perform the tasks your job demands?”

### 2.3. Statistical Analysis

First, differences among the respondents were examined by applying chi-squared tests. Next, to estimate the association and excess risk between SHS and psychosocial risk factors, odds ratios (ORs) and their corresponding 95% confidence intervals (95% CIs) were calculated and stratified by socio-demographic, employment, and labor variables, applying the Cochran–Mantel–Haenszel test to confirm or refute explanatory differences. All analyses were executed using SPSS version 21.0 (IBM Corporation, Armonk, NY, USA).

## 3. Results

Table 1 presents the general characteristics and self-perceived health of the sample. Overall, 12% of the respondents reported self-perceived ill health, which did not differ significantly by gender (*p* > 0.05). Self-perceived ill health increased progressively with age and lower educational level (*p* < 0.05).

Turning to labor characteristics, workers in the public sector (*p* < 0.05), in construction and industry (*p* > 0.05), and in micro and large enterprises (*p* > 0.05) showed a greater tendency toward self-perceived ill health. Finally, in terms of employment characteristics, workers in lower categories (unskilled jobs), those with one year or more of tenure in the current enterprise (*p* > 0.05), and those with working weeks exceeding 40 h (*p* < 0.05) reported a more significant presence of self-perceived ill health.

In general, it can be observed that the most common factors for exposure to psychosocial risks noted by the surveyed workers were cognitive demand (DGOG) and job insecurity (IL), without statistically significant differences (*p* > 0.05) by socio-demographic (Figure 1), occupational (Figure 2), or employment characteristics (Figure 3).

Figure 4 presents results concerning the association and excess risk analysis between SHS and psychosocial risk factors. Among the dimensions of psychosocial risk at work and in self-perceived ill-health were quantitative demand (DCUA; OR = 3.03; 95% CI = 1.97–4.66), social support from colleagues (ASc; OR = 2.04; 95% CI = 1.36–3.06) and bosses (ASj; OR = 1.91; 95% CI = 1.30–2.81), and finally emotional demand (DEMO; OR = 1.46; 95% CI = 1.00–2.12).

Results of analyses stratified by socio-demographic, labor, and employment characteristics are shown in Table 2. Quantitative demand (DCUA) was significantly more likely to lead to poor self-perceived health among women (OR = 3.98; 95% CI = 2.16–7.33; *p* < 0.001) and men (OR = 2.33; 95% CI = 1.26–4.30; *p* < 0.01), those aged 25–54 years (OR = 3.47; 95% CI = 2.18–5.51; *p* < 0.001), those with higher (OR = 5.73; 95% CI = 2.97–11.06; *p* < 0.001) and middle (OR = 2.19; 95% CI = 1.14–4.19; *p* < 0.01) levels of education, those in both the private (OR = 2.70; 95% CI = 1.61–4.53; *p* < 0.001) and public (OR = 3.93; 95% CI = 1.74–8.88; *p* < 0.001) sectors, those in industrial activities (OR = 8.69; 95% CI = 2.11–35.87; *p* < 0.001) and services (OR = 2.83; 95% CI = 1.73–4.61; *p* < 0.05), those in the average employment category (OR = 3.70; 95% CI = 2.22–6.18; *p* < 0.001), and with a slightly higher probability in those salaried workers who reported working ≤40 h per week (OR = 3.31; 95% CI = 1.94–5.65; *p* < 0.001) compared to >40 h per week (OR = 2.58; 95% CI = 1.24–5.37; *p* < 0.01), with significant differences observed in the Cochran–Mantel–Haenszel test (s1, *p* = 0.001).

The lack of peer social support (ASc) that was most associated with poor self-perceived health was observed among women (OR = 2.06; 95% CI = 1.16–3.66; *p* < 0.01), men (OR = 2.04; 95% CI = 1.15–3.62; *p* < 0.01), and respondents aged 25–54 years (OR = 2.23; 95% CI = 1.43–3.48; *p* < 0.001), with significant differences observed in the Cochran–Mantel–Haenszel test (s1; *p* = 0.001). Similarly, this was also the case for those salaried workers with basic education (OR = 2.84; 95% CI = 1.08–7.46; *p* < 0.05) with respect to other educational levels (s3; *p* = 0.003). In terms of labor characteristics, those working in the private sector (OR = 2.04; 95% CI = 1.26–3.28; *p* < 0.01), services (OR = 2.18; 95% CI = 1.37–3.47; *p* < 0.001), and in small enterprises (OR = 2.05; 95% CI = 1.07–3.92; *p* < 0.05) had a higher probability of poor SHS due to lack of peer social support (s1; *p* = 0.001). Workers in lower employment categories (OR = 2.29; 95% CI = 1.01–5.18; *p* < 0.05) and medium employment categories (OR = 1.86; 95% CI = 1.11–3.10; *p* < 0.01), those with 1 year or more of employment in the current company (OR = 2.05; 95% CI = 1.36–3.08; *p* < 0.001), and those who declared to work ≤40 h per week (OR = 3.02; 95% CI = 11.58–5.77; *p* < 0.001) also presented excess risk (s1 = 0.001).

Among the characteristics that were most associated with poor self-perceived health and the lack of social support from employers was being female (OR = 2.24; 95% CI = 1.29–3.89; *p* < 0.01) and being between 25 and 54 years of age (OR = 2.24; 95% CI = 1.47–3.41; *p* < 0.001), with significant differences observed in the Cochran–Mantel–Haenszel test (s1, *p* = 0.001). Salaried workers with basic educational levels (OR = 2.84; 95% CI = 1.08–7.46; *p* < 0.05) presented a higher probability of poor SHS due to lack of support from employers with respect to other educational levels (s3 = 0.003). Similarly, this was also the case in those workers employed in the private sector (OR = 2.39; 95% CI = 1.52–3.74; *p* < 0.001), services (OR = 1.68; 95% CI = 1.08–2.64; *p* < 0.01), and in medium-sized companies (OR = 2.83; 95% CI = 1.31–6.10; *p* < 0.01), with statistically significant differences (s1 = 0.001). In terms of employment characteristics, those working in lower occupations (OR = 3.35; 95% CI = 1.50–7.48; *p* < 0.01) and medium occupations (OR = 1.72; 95% CI = 1.06–2.79; *p* < 0.05) also presented statistically significant differences in this respect (s3 = 0.003). Moreover, a length of service of 1 year or more in the current company (OR = 1.91; 95% CI = 1.30–2.82; *p* < 0.001) was also associated with a greater risk of presenting poor self-perceived health, particularly where those workers declared to work ≤40 h per week (OR = 2.27; 95% CI = 11.22–4.24; *p* < 0.01) compared to those who worked ≤40 h per week (OR = 1.67; 95% CI = 1.01–2.74; *p* < 0.05), with statistically significant differences observed (s2 = 0.002).

Finally, emotional demand (DEMO) was associated with a significantly higher probability of having poor self-perceived health in those salaried workers with higher education levels (OR = 1.88; 95% CI = 1.03–3.42; *p* < 0.05), those in the higher employment category (OR = 3.46; 95% CI = 1.08–11.10; *p* < 0.05), those with a tenure of 1 year or more in the current company (OR = 1.50; 95% CI = 1.03–2.19; *p* < 0.05), and those who declared to work ≤40 h per week (OR = 1.72; 95% CI = 1.07–2.74; *p* < 0.05), but without a statistically significant difference in the Cochran–Mantel–Haenszel test (s4 ≥ 0.01).

## 4. Discussion

This paper explored the influence of exposure to psychosocial risk factors on the probability of presenting poor self-perceived health. This was accomplished through a stratified analysis of socio-demographic, labor, and employment determinants in a large sample of salaried workers in the Ecuadorian city of Guayaquil (*n* = 1049).

The findings were similar between women and men. However, several studies have observed that a lack of social support from colleagues and particularly from employers, as well as high quantitative demands at work, may affect SHS among women to a greater extent [26,27,28,29,30]. In this sense, it would be useful for organizations in general and human resources personnel more specifically to design strategies aiming to strengthen collaboration in the workplace.

In our analyses, no significant relationship was found between exposure to psychosocial risk factors and the likelihood of self-perceived ill health for the vast majority of the occupational variables analyzed, except for the influence of peer and managerial support in private and service enterprises [31].

With regard to the tenure of workers with one year or more of employment, the findings were consistent with other studies [1,29]. Length of service in the company, understood as exposure to psychosocial risk factors, may be associated with work-related stress and, therefore, the manifestation of different physical, mental, and psychosomatic symptoms in workers [15,16,17,18].

Finally, in our results, we observed that workers with long working hours (>40 h per week) who suffered from a lack of social support (bosses and colleagues) and high demands or requirements in their work activities (emotional, cognitive, and quantitative) had a higher risk of self-perceived ill health [17,18,19]. It has hitherto been shown that workers exposed to these psychosocial risk factors have higher rates of many different illnesses [32,33,34]. It is therefore important to devise initiatives that improve the work climate, minimize work–family conflicts, and make workloads and hours more flexible in an attempt to counteract negative health effects.

It is necessary to take into account some limitations of the study. This was a cross-sectional analysis based on data from 2017, carried out in a specific geographical area, and in a particular working population, i.e., the formal sector. However, by virtue of the sample design, it is reasonable to suggest that the results reflect the population they represent in Ecuador’s second-largest city. This study could provide a useful starting point and framework for cognate analyses that configure longitudinal research designs, nationwide research designs, or both. In addition, the use of the self-perceived health indicator could be another limitation, as it is a subjective measure. However, in similar studies, it has been shown to be valid and reliable [2]. While the generalization of these results should be approached with due caution, the findings presented allow for a better understanding of the relationship between exposure to psychosocial risks and poor self-perceived health. They can usefully be compared to related extant studies [21,23].

## 5. Conclusions

In conclusion, the results of this study for salaried workers in the Ecuadorian city of Guayaquil reveal a multitude of psychosocial risk factors present in the workplace. These may cause damage to health for the coming years. This evidence demonstrates the need to design interventions vis-à-vis psychosocial risks that will reduce damage to workers’ health and bring about economic, social, and organizational improvements for companies.

Further, to continue generating knowledge in this research domain, it would be advisable to explore data spanning other geographical areas to verify or refute the trends identified and explored herein. In this sense, an appropriate strategy would be to set up multidisciplinary working groups that could implement the same survey and statistical methodology. Finally, public bodies need to reinforce the importance of psychosocial risks in a more specific way to ensure the health and well-being of employees.

## Figures and Tables

**Figure 1 ijerph-17-09099-f001:**
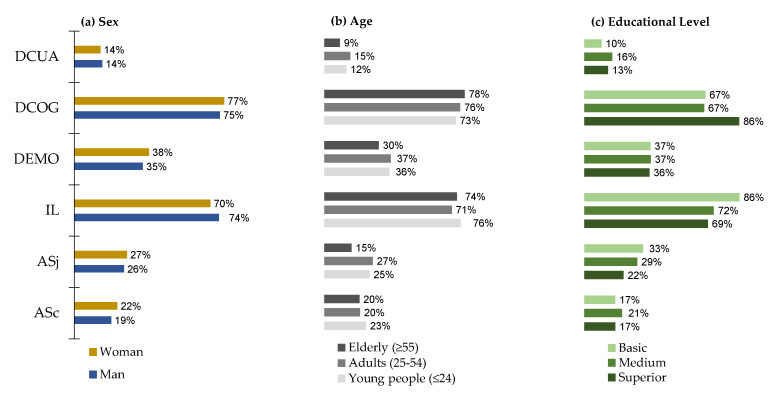
Psychosocial risk factors by socio-demographic characteristics (*n* = 1049). ASc = social support from colleagues (% answered No), ASj = social support from bosses (% answered No), IL = job insecurity (% answered Yes), DEMO = emotional demand (% answered Yes/Partly), DCOG = cognitive demand (% answered Yes/Partly), DCUA = quantitative demand (% answered No/Partly).

**Figure 2 ijerph-17-09099-f002:**
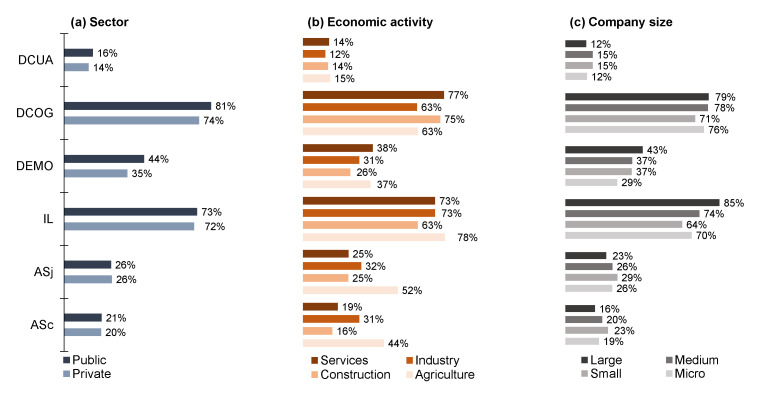
Psychosocial risk factors by job characteristics (*n* = 1049). ASc = social support from colleagues (% answered No), ASj = social support from bosses (% answered No), IL = job insecurity (% answered Yes), DEMO = emotional demand (% answered Yes/Partly), DCOG = cognitive demand (% answered Yes/Partly), DCUA = quantitative demand (% answered No/Partly).

**Figure 3 ijerph-17-09099-f003:**
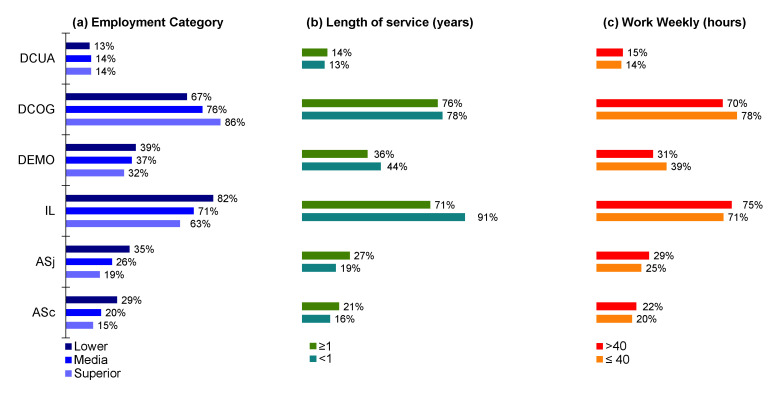
Psychosocial risk factors by employment characteristics (*n* = 1049). ASc = social support from colleagues (% answered No), ASj = social support from bosses (% answered No), IL = job insecurity (% answered Yes), DEMO = emotional demand (% answered Yes/Partly), DCOG = cognitive demand (% answered Yes/Partly), DCUA = quantitative demand (% answered No/Partly).

**Figure 4 ijerph-17-09099-f004:**
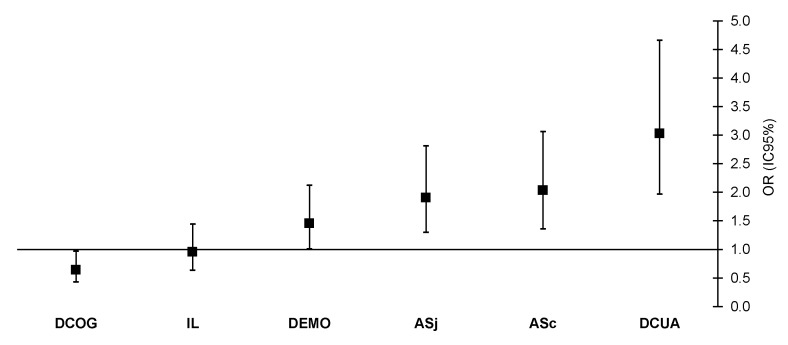
Odds Ratios of psychosocial risk factors and poor SHS. ASc = social support from colleagues, ASj = social support from bosses, IL = job insecurity, DEMO = emotional demand, DCOG = cognitive demand, DCUA = quantitative demand.

**Table 1 ijerph-17-09099-t001:** General characteristics and self-perceived health (SHS) of the sample (*n* = 1049).

Variables	*n*	%*n*	SHS		*p*-Value
			Good	Poor	
**Socio-Demographics**					
**Sex**					
Women	519	49.5	88.2	11.8	0.660
Men	530	50.5	87.4	12.6	
**Age (years)**					
Young people (≤24)	146	13.9	93.2	6.8	0.002
Adults (25–54)	857	81.7	87.6	12.4	
Elderly (≥55)	46	4.4	73.9	26.1	
**Educational Level**					
Superior	477	45.5	89.9	10.1	0.001
Medium	477	45.5	88.1	11.9	
Basic	95	9.1	75.8	24.2	
**Labor**					
**Sector**					
Private	840	80.1	89.3	10.7	0.003
Public	209	19.9	81.8	18.2	
**Economic Activity**					
Agriculture	27	2.6	88.9	11.1	0.511
Construction	81	7.7	82.7	17.3	
Industry	91	8.7	86.8	13.2	
Services	850	81	88.4	11.6	
**Company Size**					
Micro	151	14.4	85.4	14.6	0.402
Small	384	36.6	87.8	12.2	
Medium	305	29.1	90.2	9.8	
Large	209	19.9	86.1	13.9	
**Employment**					
**Employment Category**					
Superior	93	8.9	84.9	15.1	0.000
Medium	812	77.4	90.0	10.0	
Lower	144	13.7	77.1	22.9	
**Tenure (years)**					
<1	32	3.1	96.9	3.1	0.111
≥1	1017	96.9	87.5	12.5	
**Weekly Work (hours)**					
≤40	731	69.7	89.2	10.8	0.036
>40	318	30.3	84.6	15.4	

**Table 2 ijerph-17-09099-t002:** Stratified odds ratios of psychosocial risk factors at work and poor SHS.

Variables	ASc		ASj		IL	
	OR (95% CI)	Test ^a^	OR (95% CI)	Test ^a^	OR (95% CI)	Test ^a^
**Socio-Demographics**						
**Sex**						
Women	2.06 (1.16–3.66) **	s1	2.24 (1.29–3.89) **	s1	1.14 (0.63–2.06)	s4
Men	2.04 (1.15–3.62) **		1.64 (0.95–2.83)		0.80 (0.46–1.41)	
**Age (years)**						
Young people (≤24)	1.45 (0.35–5.95)	s1	0.72 (0.15–3.56)	s1	1.28 (0.26–6.34)	s4
Adults (25–54)	2.23 (1.43–3.48) ***		2.24 (1.47–3.41) ***		0.98 (0.63–1.54)	
Elderly (≥55)	1.56 (0.32–7.52)		1.16 (0.19–6.95)		0.62 (0.15–2.59)	
**Educational Level**						
Superior	1.94 (0.98–3.85) *	s3	1.55 (0.80–3.01)	s3	0.81 (0.43–1.52)	s4
Medium	1.56 (0.84–2.92)		2.07 (1.18–3.65) **		1.09 (0.59–2.05)	
Basic	2.84 (1.08–7.46) *		1.87 (0.71–4.92)		0.45 (0.13–1.55)	
**Labor**						
**Sector**						
Private	2.04 (1.26–3.28) **	s1	2.39 (1.52–3.74) ***	s1	1.18 (0.72–1.94)	s4
Public	2.08 (0.95–4.58)		1.03 (0.46–2.29)		0.56 (0.27–1.18)	
**Economic Activity**						
Agriculture	2.80 (0.22–35.29)	s1	1.27 (0.97–1.67)	s1	-	s4
Construction	1.56 (0.37–6.58)		2.84 (0.85–9.53)		1.07 (0.32–3.56)	
Industry	1.74 (0.50–6.05)		2.44 (0.71–8.34)		0.48 (0.14–1.66)	
Services	2.18 (1.37–3.47) ***		1.68 (1.08–2.64) **		1.02 (0.63–1.62)	
**Company Size**						
Micro	2.40 (0.87–6.60)	s1	1.81 (0.69–4.71)	s1	0.73 (0.28–1.89)	s4
Small	2.05 (1.07–3.92) *		1.83 (0.98–3.45)		0.88 (0.47–1.65)	
Medium	1.80 (0.78–4.14)		2.83 (1.31–6.10) **		1.47 (0.58–3.75)	
Large	2.26 (0.90–5.63)		1.38 (0.57–3.35)		0.85 (0.30–2.41)	
**Employment**						
**Employment Category**						
Superior	1.69 (0.41–7.02)	s1	0.33 (0.04–2.71)	s3	1.04 (0.32–3.42)	s4
Medium	1.86 (1.11–3.10) **		1.72 (1.06–2.79) *		0.90 (0.55–1.49)	
Lower	2.29 (1.01–5.18) *		3.35 (1.50–7.48) **		0.77 (0.29–2.02)	
**Tenure (years)**						
<1	0.96 (0.89–1.04)	s1	-	s2	1.04 (0.97–1.11)	s4
≥1	2.05 (1.36–3.08) ***		1.91 (1.30–2.82) ***		0.98 (0.65–1.47)	
**Weekly Work (hours)**						
≤40	1.56 (0.91–2.66)	s1	1.67 (1.01–2.74) *	s2	1.37 (0.80–2.35)	s4
>40	3.02 (1.58–5.77) ***		2.27 (1.22–4.24) **		0.51 (0.26–0.97)	
**Socio-Demographics**						
**Sex**						
Women	1.49 (0.88–2.50)	s4	0.69 (0.38–1.25)	s4	3.98 (2.16–7.33) ***	s1
Men	1.44 (0.84–2.46)		0.61 (0.35–1.06)		2.33 (1.26–4.30) **	
**Age (years)**						
Young people (≤24)	1.83 (0.51–6.65)	s4	0.84 (0.21–3.42)	s4	1.88 (0.37–9.62)	s1
Adults (25–54)	1.42 (0.94–2.14)		0.60 (0.39–0.94) *		3.47 (2.18–5.51) ***	
Elderly (≥55)	1.98 (0.50–7.87)		0.78 (0.17–3.66)		0.94 (0.09–10.00)	
**Educational Level**						
Superior	1.88 (1.03–3.42) *	s4	0.66 (0.31–1.44)	s4	5.73 (2.97–11.06) ***	s1
Medium	1.09 (0.62–1.92)		0.71 (0.40–1.25)		2.19 (1.14–4.19) **	
Basic	1.83 (0.71–4.76)		0.68 (0.26–1.82)		1.65 (0.38–7.20)	
**Labor**						
**Sector**						
Private	1.43 (0.92–2.23)	s4	0.66 (0.41–1.05)	s4	2.70 (1.61–4.53) ***	s1
Public	1.34 (0.66–2.72)		0.50 (0.22–1.12)		3.93 (1.74–8.88) ***	
**Economic Activity**						
Agriculture	4.00 (0.31–51.03)	s4	0.25 (0.02–3.19)	s4	3.50 (0.24–51.46)	s1
Construction	2.60 (0.78–8.67)		0.52 (0.15–1.79)		2.01 (0.46–8.79)	
Industry	1.15 (0.32–4.17)		1.22 (0.34–4.42)		8.69 (2.11–35.87) ***	
Services	1.40 (0.92–2.14)		0.63 (0.40–1.01)		2.83 (1.73–4.61) ***	
**Company Size**						
Micro	0.93 (0.34–2.57)	s4	0.30 (0.12–0.78) *	s4	2.62 (0.83–8.29)	s1
Small	1.21 (0.65–2.26)		0.84 (0.43–1.62)		3.63 (1.83–7.19) ***	
Medium	2.13 (1.00–4.55)		0.61 (0.27–1.40)		4.73 (2.09–10.66) ***	
Large	2.13 (1.00–4.55)		0.81 (0.32–2.05)		1.28 (0.40–4.06)	
**Employment**						
**Employment Category**						
Superior	3.46 (1.08–11.10) *	s4	0.97 (0.19–4.94)	s4	3.11 (0.81–12.02)	s1
Medium	1.35 (0.85–2.16)		0.57 (0.35–0.92) *		3.70 (2.22–6.18) ***	
Lower	1.21 (0.55–2.67)		0.96 (0.42–2.19)		1.68 (0.58–4.82)	
**Tenure (years)**						
<1	0.94 (0.84–1.06)	s4	-	s4	1.33 (0.76–2.35) **	s1
≥1	1.50 (1.03–2.19) *		0.67 (0.45–1.00) *		2.93 (1.89–4.53) ***	
**Weekly Work (hours)**						
≤40	1.72 (1.07–2.74) *	s4	0.70 (0.42–1.19)	s4	3.31 (1.94–5.65) ***	s1
>40	1.19 (0.63–2.26)		0.62 (0.33–1.17)		2.58 (1.24–5.37) **	

ASc = social support from colleagues, ASj = social support from bosses, IL = job insecurity, DEMO = emotional demand, DCOG = cognitive demand, DCUA = quantitative demand. * *p* < 0.05; ** *p* < 0.01; *** *p* < 0.001. ^a^ Cochran–Mantel–Haenszel test; s1 = 0.001; s2 = 0.002; s3 = 0.003; s4 = >0.01.

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
