# Peer review of "Salaried Workers’ Self-Perceived Health and Psychosocial Risk in Guayaquil, Ecuador"

_ijerph, 2020, doi:10.3390/ijerph17239099_

Round 1

Reviewer 1 Report

The article analyzes the relationships between self-perceived health and psychosocial risk in Guayaquil, using elements of a survey conducted in 2017.
The introduction can be better justified. from the references presented in the introduction, only 4 have less than 5 years and only 3 are less than 3 years.
The conclusions presented can also be further developed and deepened. From the two paragraphs of the conclusions, the first gives a generic description, not focused on the results and the second gives a short overview of the research needs.

Keywords are, in general, repetitions of words in the title, not adding information to the article. The keywords "Determining factors" and "Workplaces", never appear during the text.

In lines 53 and 54, the authors should confirm that what they meant is "As discussed above, the detrimental effects of psychosocial health risks are well documented in the workplace.", Since it does not seem to make sense. It can be a translation problem.

The graphs in figure 1, in b), are missing legends referring to age. it only has subtitles for > = 55.

The graphs in figure 3, in s), are missing legends referring to age. it only has subtitles for > 40.

In lines 223 and 233, the authors use the first person (our and we). they must reformulate the sentences since in scientific language the 1st person should be avoided.

Author Response

We want to thank you for the valuable comments and suggestions made in our manuscript. We believe that these comments have helped us to introduce changes that substantially improve the document. The authors of the manuscript have carefully reviewed all comments and suggestions and made a significant effort to provide point-by-point answers to clarify and overcome the reviewers' concerns. Our answers and comments are in red font.

The introduction can be better justified from the references presented in the introduction, only 4 have less than 5 years and only 3 are less than 3 years.

Thanks for the recommendation. We have included three actualized references to enhance the introduction [8,9,10].  These studies are more current and reflect the problems of our study with greater precision

Point 2: The conclusions presented can also be further developed and deepened. From the two paragraphs of the conclusions, the first gives a generic description, not focused on the results and the second gives a short overview of the research needs.

We appreciate this observation. We consider, in our opinion, as this is the first study that analyzes this problem in the country, the conclusions included are, on the one hand, to reflect this problem and, on the other hand, a call for public decision-making by the responsible bodies in the formulation of policies. We would appreciate this consideration.

"In conclusion, the results of this study for salaried workers in the Ecuadorian city of Guayaquil reveal a multitude of psychosocial risk factors present in the workplace. These may cause damage to health for the coming years. This evidence demonstrates the need to design interventions vis-à-vis psychosocial risks that will reduce damage to workers' health and bring about economic, social, and organizational improvements for companies.

Further, with the intention of continuing to generate knowledge in this domain of research, it would be advisable to explore data spanning other geographical areas, in order to verify/refute the trends identified and explored herein. In this sense, an appropriate strategy would be to set up multidisciplinary working groups, using the same survey and statistical methodologies. Finally, it is important for public bodies to reinforce the importance of psychosocial risks in a more specific way with a view to ensuring the health and wellbeing of employees."

Point 3: Keywords are, in general, repetitions of words in the title, not adding information to the article. The keywords "Determining factors" and "Workplaces", never appear during the text.

We appreciate this observation. We have considered modifying the two keywords; Determining factors by Working conditions and Workplaces by Salaried workers. We consider that these keywords better fit the study presented.

Point 4: In lines 53 and 54, the authors should confirm that what they meant is "As discussed above, the detrimental effects of psychosocial health risks are well documented in the workplace.", Since it does not seem to make sense. It can be a translation problem.

Thanks for this observation. We have improved the wording of the paragraph.

"In particular, the Republic of Ecuador is among the unequal countries in Latin America and the Caribbean [20,21] and, to our knowledge, there are no studies on the detrimental effects of psychosocial risks on the health of workers in the country, although they have been widely discussed in other contexts [7]. In the absence of evidence on this reality [16], it is important to generate knowledge about this phenomenon for decision-making and public interventions based on evidence to improve working conditions [22]."

Point 5: The graphs in figure 1, in b), are missing legends referring to age. it only has subtitles for > = 55.

We appreciate your observation of the captions in figure 1, b). The chart legend has been corrected, inclement; adults and young people.

Point 6: The graphs in figure 3, in s), are missing legends referring to age. it only has subtitles for > 40.

Thanks, we have resolved the error in figure 3, c) Work Weekly (hours).

Point 7: In lines 223 and 233, the authors use the first person (our and we). they must reformulate the sentences since in scientific language the 1st person should be avoided.

The sentences have been reformulated. We appreciated the observation.

Reviewer 2 Report

The objectives of the study are not described in my opinion with sufficient clarity. It seems that the work has two complementary objectives (relationship between self-perceived health and occupational stressors, on the one hand, and association between the occupational stressors and work and demographic characteristics, on the other hand). 

I consider that a deep rethinking of the study design is necessary, which could include the substitution of the stratified analysis for a multivariate one, performing the analysis of the relationship between perceived health (dependent variable) and psychosocial risk factors, including the explanatory factors of interest in the analysis.

All of this should lead to a complete reworking of the discussion and conclusions sections according to the results obtained after the new analysis.

There are some additional comments on the current versión of your manuscript:

As a result of the chosen analysis strategy, the presentation of the data is prolix and unclear.

In table 1, the chi square test should be done for each value of the variable (in relation to the rest of the subjects). In the variables where it is appropriate (age, company size, job category), a chi square of trends should also be performed, necessary to justify comments such as the one of lines 107-111.

There is a mistake in the acronym of the variable DCUA in line 138.

Text of lines 233-236 of the current manuscript is not supported by data shown in the report.

Finally, the representativeness of the results of an study, and its external validity, do not rely only on the size of the sample. Additional criteria must be taken into account.

Author Response

We want to thank you for the valuable comments and suggestions made in our manuscript. We believe that these comments have helped us to introduce changes that substantially improve the document. The authors of the manuscript have carefully reviewed all comments and suggestions and made a significant effort to provide point-by-point answers to clarify and overcome the reviewers' concerns. Our answers and comments are in red font.

1: The objectives of the study are not described in my opinion with sufficient clarity. It seems that the work has two complementary objectives (relationship between self-perceived health and occupational stressors, on the one hand, and association between the occupational stressors and work and demographic characteristics, on the other hand).

We appreciate this observation; we have modified the study objective's wording in line 60.

"The objective of this study was to determine the relationship between self-perceived poor health and exposure to psychosocial risk factors by sociodemographic, labour, and employment characteristics."

2: I consider that a deep rethinking of the study design is necessary, which could include the substitution of the stratified analysis for a multivariate one, performing the analysis of the relationship between perceived health (dependent variable) and psychosocial risk factors, including the explanatory factors of interest in the analysis. All of this should lead to a complete reworking of the discussion and conclusions sections according to the results obtained after the new analysis.

We appreciate this comment. We have enhanced the study objective for better clarity. A future study would be to carry out a multivariate logistic regression to estimate the relationship between the dependent variable (health status) and independent variables (psychosocial risk factors) and demographic, labour and employment characteristics as covariates. We believe that this analysis will provide greater precision to the study.

3: As a result of the chosen analysis strategy, the data's presentation is prolix and unclear.

We appreciate this comment. We have enhanced the study objective for better clarity. A future study would be to carry out a multivariate logistic regression to estimate the relationship between the dependent variable (health status) and independent variables (psychosocial risk factors) and demographic, labour and employment characteristics as covariates. We believe that this analysis will provide greater precision to the study.

4: In table 1, the chi square test should be done for each value of the variable (in relation to the rest of the subjects). In the variables where it is appropriate (age, company size, job category), a chi square of trends should also be performed, necessary to justify comments such as the one of lines 107-111.

The variables included in the applied survey (I-ECSST) have closed categories of responses, particularly the variables age, company size, etc. This limitation prevents us from performing the chi-square test.

5: There is a mistake in the acronym of the variable DCUA in line 138.

We appreciate this observation. We have corrected the error in the acronym of the variable DCUA on line 138.

6: Text of lines 233-236 of the current manuscript is not supported by data shown in the report.

We appreciate the very timely recommendation. We have included three bibliographic references that support our findings with other studies. References 17,18 and 19.

Finally,  results showed that workers with long working hours (>40 hours per week), who suffer from a lack of social support (bosses and colleagues) and high demands or requirements in their work activities (emotional, cognitive, and quantitative) have a higher risk of self-perceived ill health [17,18,19].

Point 7:  Finally, the representativeness of the results of a study, and its external validity, do not rely only on the size of the sample. Additional criteria must be taken into account.

We appreciate this observation; we have modified the text of line 244.

"However, by virtue of the sample design, it is reasonable to suggest that the results reflect the population they represent in Ecuador's second-largest city."